# Older Adults’ Demand for Community-Based Adult Services (CBAS) Integrated with Medical Care and Its Influencing Factors: A Pilot Qualitative Study in China

**DOI:** 10.3390/ijerph192214869

**Published:** 2022-11-11

**Authors:** Zhenyu Wang, Hanchun Wei, Zhihan Liu

**Affiliations:** 1Center for Chinese Public Administration Research, School of Government, Sun Yat-sen University, Guangzhou 510006, China; 2School of Public Administration, Central South University, Changsha 410075, China

**Keywords:** community-based adult services, integrated care, older people, needs, influencing factors, qualitative

## Abstract

Introduction: The number of older people in China who require formal care is increasing. In response, China is creating a service delivery mode of health and social care combination for older people—community-based adult services (CBAS) integrated with medical care—in some provincial capital cities, such as Changsha. However, the needs of most older people for this service delivery mode are not well understood. Aim: To assess older people’s awareness of and demand for CBAS integrated with medical care and to determine influencing factors. Methods: Semi-structured guideline interviews were conducted with 20 older people (aged 65+ years) from two communities at different economic development levels and from a nursing home in Changsha, China. Interviews were analyzed using qualitative content analysis. Results: The specific needs that older adults expect from CBAS integrated with medical care involve daily care, primary care, self-management guidance, rehabilitation therapy services, and mental health services. Contrary to expectations, most interviewees showed low awareness of and demand for CBAS integrated with medical care. Individual, family, and community factors influence older people’s demand, as do exogenous variables such as gender and number of children. Discussion: The influencing mechanism of older people’s demand for CBAS integrated with medical care is complex and multifaceted. To implement and promote CBAS integrated with medical care, attention should be given to older people’s individual needs, family backgrounds and community environment improvement. Furthermore, improving awareness of integrated care and increasing ageing-in-place opportunities for more older adults is essential and urgent.

## 1. Introduction

China is witnessing the rapid growth of ageing [1]. More than 50% of people aged 75 or older have multiple chronic diseases [2], realizing the goal of “aging with quality care, aging with responsive support” has become increasingly challenging. The community and family are the basic living units for human beings, thus, integrated care programs are becoming increasingly popular as a way of meeting the nursing demand of older adults to age in place [3]. As part of ageing-in-place policies and integrated care programs, community-based services for older adults are designed to offer a range of services, including personal care, nutrition and meal services, and transportation services [4]. Accepting older adult services integrated with medical care not only allows older adults to stay with their family and friends [5] but also reduces care expenditures for older adults by integrating family support and organizational services [6]. Most importantly, it dramatically improves older adults’ quality of life [7]. Although community-based care services have many advantages, older adults’ acceptance of them remains low [8]. For example, research on adult day care has found that many older people and caregivers who could benefit do not use these services [9]. Therefore, it is worth exploring what factors influence the demand for community-based care services for older people.

Conversely, the implementation of integrating health care and social care for older adults in China is a recent development. Since 2013, the State Council has promulgated “Several Opinions on Accelerating the Development of the Aged Care Service Industry”, proposing to add medical care service to traditional aged care service and develop a healthy old-age care service industry named *“Yiyang Jiehe”* in Chinese [10]. In June 2016, the Health and Family Planning Commission and the Ministry of Civil Affairs issued the “Notice on Determining the First Batch of Pilot Units for the Integration of Medical and Older Adults Care at the National Level”, identifying 50 regions as the first batch of pilot units for the integration of medical and older adults care in the country. In September of the same year, the Health and Family Planning Commission and the Ministry of Civil Affairs released the list of the second batch of pilot units. As of 2017, China has set up 90 national-level pilot cities for the implementation of “*Yiyang Jiehe*”.

After nearly a decade of development, four primary modes of integrating medical care and ageing services have been formed in China. The first mode is “constructing hospital in nursing home”: a professional medical care team is set up in the nursing home to integrate the older adults care and health care functions [11]. The second mode is “nursing in hospital”, including two pathways: establishing geriatric departments in some large hospitals to provide medical, nursing, rehabilitation, hospice, and other services for older people; transforming some primary care institutions with limited resources into integrated health and social care institutions [12]. The third model is “contract cooperation between hospitals and nursing homes”. On the one hand, hospitals will regularly carry out basic diagnosis and treatment services such as common disease detection, chronic disease and geriatric disease management, and health education for older people in nursing homes. On the other hand, the hospital also provides timely and convenient medical referral services for the older adults in need [13]. The fourth mode is “community-based adult services (CBAS) integrated with medical care”, which offers aged care services in the home- and community-based settings under the direction of the government, so as to realize the combination of medical care, hierarchical diagnosis and treatment, resource sharing and service collaboration.

As the fourth mode of China’s integrated care and a move to defragmentation of aged care services in community settings, CBAS integrated with medical care aims for a “person-centered” approach that connects the healthcare system with other human service systems for older adults [10]. In this mode, older adults can age in place in a family-like environment based in a residential community, with multiple demands being satisfied. The description of the CBAS integrated with medical care is shown in Table 1. Its provision has been relatively well-established in many developed countries, such as the PACE [14] and CBSS [15] programs in the USA, the social and community care of NHS [16] in the UK, long-term care services [17] in Japan and the “Embrace” service program [18] in The Netherlands.

To improve the quality of life and happiness of older adults, it is essential to understand older adults’ awareness of, attitudes towards, and factors influencing their demand for services [23]. However, previous research on eldercare services integrated with medical care has mainly focused on the connotations of, problems with, and methods of implementing the model but has seldom paid attention to older adults’ awareness of and willingness to participate in the CBAS integrated with medical care healthcare model [24,25]. The health and care needs of older populations with chronic diseases are not well-recognized, known, or met [26], and the unsatisfied needs include those related to medical care, mental health, nutrition, exercise, and other domains [27]. Furthermore, little is known about the factors that influence older people’s and families’ decisions to utilize these care services [28]. Most scholars in China rely on quantitative research methods [29,30], and they seldom perform qualitative studies to thoroughly explore older adults’ real demand for CBAS integrated with medical care.

Due to the lack of previous studies, we conducted a pilot qualitative study to answer the following research questions: 1. How well do older people currently know about CBAS integrated with medical care and what services do older people need? 2. What factors influence the demand for CBAS integrated with medical care among older adults? Furthermore, we hope that this pilot study will contribute to promoting CBAS integrated with the medical care model across the country and will provide a reference for decision-making to accelerate the development of ageing services in China.

## 2. Methods

A qualitative descriptive design [31] was adopted. The data were collected through interviews [32] and were subsequently analyzed via inductive content analysis [33].

### 2.1. Study Design and Setting

Hunan province is one of the most aging provinces in south central China, and Changsha (the capital city of Hunan Province) is a pilot city experimenting with older adult services integrated with medical care, with the financial support of the central government. There are 272 service centers for CBAS integrated with medical care [34]. The target group in this study was older adults in Changsha, Hunan. The older adults we recruited came from two communities at different levels of economic development and a nursing home; this improved the representativeness of the sample. The research was conducted in J community (most residents here are well educated or government officials who have higher pensions and economic capability), R community (the residents who moved here from rural areas after their land was expropriated by the government, generally have only modest pensions and lower economic capability), and W nursing home. The research was conducted at a nursing home for the following reasons: (1) some older residents in nursing home did not move there voluntarily or did not move with a strong desire to do so, and (2) older residents in nursing home have differing opinions and experiences than those who age in place. The recruitment requirements were as follows: (1) 60 years old or older, (2) no significant hearing or vision impairments, and (3) able to clearly express personal feelings and thoughts independently. The selection of participants with different experiences increases the possibility of shedding light on the research question from a variety of aspects and enhances credibility [35].

### 2.2. Sample

The sample included 8 men and 12 women, fourteen of the participants had partners, and the other six were widowed. The mean age was 74 years (65–89 years). All participants were from Hunan, China. Five participants’ monthly incomes were less than RMB 1000 (approximately USD 143), six received RMB 2500–4000 (approximately USD 357–571) per month, five received RMB 4000 to 5500 (approximately USD 571–786) per month, and four received over RMB 5500 (approximately USD 786) per month. Seventeen of the participants were covered by social security healthcare insurance, and three were not covered. Six participants had one child, four had one son and one daughter, six had two sons or two daughters, three had one son and two daughters, and one had no children. The sample characteristics of the 20 interviewees are shown in Table 2.

### 2.3. Data Collection

The data collection has been approved by the Clinical Medical Ethics Committee of Xiangya Hospital, Central South University (No. 202011184). Informed consent was obtained from all respondents involved in the study. We collected information by conducting face-to-face semi-structured interviews in November and December 2020. Prior to the interviews, the study participants gave written consent to be audio-recorded. All materials were prepared in simplified Chinese. To examine the participants’ definitions and perceptions of CBAS integrated with medical care, every participant was asked six open-ended interview questions (found in the Appendix A).

The interviewers maintained a neutral stance to encourage the older adults to fully express themselves. Each interview lasted between 30 and 45 min and was conducted in the Mandarin and/or Changsha dialect. We recorded nonverbal communication signs such as tone, intonation, facial expressions, and body movement. The audio recordings of the interviews were subsequently transcribed. Respondents were recruited until data saturation was reached (no new themes occurred), which was indicated by information redundancy [36].

### 2.4. Research Trustworthiness

To improve the reliability of the study, it is necessary to demonstrate a link between the results and the data [32]. Therefore, we will describe the analysis process and results in sufficient detail to consider trustworthiness, credibility, dependability, and confirmability, and all authors checked the relevance of the results stated while carefully considering the units of meaning and clustering to finalize the classification and main idea [37]. Specifically, at first, two authors of this research team (Z.W. and H.W.) analyzed the same 20 interview transcripts in the original language (5 days apart), all themes and concepts derived from the analysis were repeatedly compared with the transcripts to ensure their validity. Links between the categories that emerged were only retained after repeated validation from the transcripts [38]. Secondly, the internal validity of content analysis can be assessed by using agreement coefficients [39]. Comparing the results of the first coding (Z.W.) with the second coding (H.W.) using the “coding comparison” function in NVivo 12.0, yielding a 95.79% coding consistency result. This indicates that the concepts and categories obtained in this study have good reliability.

### 2.5. Analysis

Because the knowledge of CBAS integrated with medical care is fragmented, an inductive approach to analysis is more appropriate [40]. Thus, we used NVivo 12.0 qualitative data analysis software to code the 20 interview transcripts and analyzed them using the qualitative content analysis method described by Elo and Kyngäs [33]. Inductive content analysis involves three steps: open coding, creating categories, and abstraction. First, the authors read the full text to obtain an overview of the data by writing notes and headings. Second, we organized the data by grouping meaning units about factors influencing the care needs of older people into broader higher order categories so that those data were similar or dissimilar. Third, grouped data (words and meaning units) with the same meaning were grouped into categories and subcategories. In the final stage of the analysis, the categories, codes, and relationships therein were provided in diagrammatic form.

After a process of reflection and discussion, we identified the factors influencing older adults’ demand to participate in CBAS integrated with medical care (as shown in Table 3).

## 3. Results

### 3.1. Specific Needs That Older Adults Expect from CBAS Integrated with Medical Care

Daily care and self-management guidance were the services that interviewees mentioned most frequently, and each of them had four tree nodes. Many older people hoped that the community would provide affordable group dining services. Primary care and rehabilitation therapy services were the second-most needed types, each with three reference nodes, followed by mental health services, which only one participant proposed.

### 3.2. Factors Influencing the Demand for CBAS Integrated with Medical Care

According to the analysis, 8 of the 20 older adults (40%) required CBAS integrated with medical care, whereas the other 12 (60%) did not. The factors affecting the needs of older people with CBAS integrated with medical care can be described by three categories: individual-level factors, family-level factors, and community-level factors (as shown in Table 4).

### 3.3. Individual-Level Factors

Five different individual-level factors were identified: (a) wellness, (b) economic capability, (c) individual opinion, (d) experiences, and (e) willingness to pay.

#### 3.3.1. Wellness

Physical condition and self-care ability determined the degree of demand for services among older people, which greatly affected their living arrangements. Older people often regarded poor physical conditions as a signal to seek CBAS integrated with medical care. Y19 described his experience as follows: “I am in good health now. I can take care of myself, can prepare food by myself. In addition, I have no chronic diseases and do not need to take medicine.” Similarly, L03 shared, “Only when we cannot walk or move will we know (whether CBAS integrated with medical care is needed).” He explained,

Before the age of 70, my body (was well with no major diseases) did not need these services at all. Currently, my wife can handle the housework and cooking well, and there is no special requirement of help from others.

However, older people with chronic diseases or poor physical conditions often require CBAS integrated with medical care. S10 complained, “I have lumbar spine pain. It will not work without taking medicine… (in the community) some check-ups and treatments are needed for older people.” This description was in stark contrast with those of Y19 and L03.

#### 3.3.2. Economic Capability

The assessment of the economic capability of older individuals was mainly based on the observation of monthly income and payment for healthcare expenses. Older people with better economic capability were more willing to accept CBAS integrated with medical care. For example, L02, who retired from the Health Bureau, had a monthly retirement salary of RMB 5000 and employee medical insurance. She said that what she most expected was CBAS integrated with medical care, and she was not willing to go to a nursing home. By contrast, older people with low retirement wages or living on state subsidies were reluctant to participate in community pension service projects. H09 stated, “I only have money from the country, approximately 300 yuan every month… (CBAS integrated with medical care) is a good project, but if you do not pay, you can’t receive services. I will insist on taking blood pressure medicine… No need (CBAS integrated with medical care), no matter how good it is, it costs money.”

#### 3.3.3. Individual Opinion

This topic involved many aspects of older people’s personal thoughts and concepts, including views on ageing, expectations for retired life, concerns for future health and food and housing, and opinions about providers of CBAS integrated with medical care. L03 espoused the traditional Chinese concept of ageing: “raise children for old age.” He believed that the responsibility for providing for him should rest on his son rather than on society. He asked us, “Why go to other places to age? My son can take care of me.” Similarly, W11 also said that when she could no longer take care of herself, her son would continue to care for her. However, in general, more people expected to grow old in the community and worried about their future health. L02 reflected, “Everyone has their own personality. It is not very good to go to a nursing home in an unfamiliar situation. Therefore, I think it is better to stay at home in the community… In addition, once you’re old, you have to think about your health problems.” Moreover, L04 also thought it was better to grow old in the community. She explained patiently,

In the case of ageing at home, the next generation is too busy to rely on them. In the case of ageing in community, the community should be well prepared to carry out activities suitable for older people… it’s the best way to meet the needs (psychological comfort) of older people in the community. We can go to the senior center during the day to have a meal and participate in activities until we go home and sleep at night.

In addition, L03, who lived at home, said, “Community will not support us.” Another older person, L13, had to live in a nursing home because the community services and infrastructure were unsatisfactory. She declared, “It is impossible to provide (CBAS integrated with medical care) in our community!” Both of these older people had a negative attitude towards providers of CBAS integrated with medical care.

#### 3.3.4. Experiences

Older people’s past and current experiences affected their needs, as did their impressions and judgements of community services for older people. Insufficient publicity of CBAS integrated with medical care, which led to an insufficient understanding among the older population, was the largest influencing factor. We found that more than half of the older adults surveyed did not understand the current status of CBAS integrated with medical care services and developments. L02, who was retired from the Health Bureau, was affected by occupation and she was the individual who knew the most about CBAS integrated with medical care in our study. Although every community in Changsha now has an activity center for older people, experience with the center is limited due to insufficient publicity. P18 pointed out, “I have lived here (in the community) for four years and just learned that there is a community service center here.” Because of insufficient publicity, he lowered his expectations of CBAS integrated with medical care. Some older people had heard negative comments about nursing homes from friends, and they were more willing to enjoy older people care services in the community after being affected. H09 said, “I have a friend. He and his wife lived in a nursing home for five or six years. After a long time, they still said that living in a nursing home was uncomfortable.”

Older people living in nursing homes described having richer experiences and ideas when they compared their original community life with their current institutional life. X08 explained why she did not continue living in the community.

First, the community environment is not good, and the air is polluted. I live here (a nursing home) with clean air. Second, living in the community, I have to do everything by myself, such as hygiene, grocery shopping, and cooking (I do not want to do these things anymore), so I just live here.

L04 was more willing to live in the community after joining a community charity organization. She emphasized, “…The organization I joined also does not advocate nursing homes because older people living there are very lonely.”

#### 3.3.5. Willingness to Pay

Older people’s willingness to pay for a certain number of consumer goods or services, such as CBAS integrated with medical care, directly affected their needs. In our study, there were more older people who were willing to pay than those who were unwilling to pay. However, L04 believed that her willingness to pay depended on the situation. She explained, “It depends on whether the value is equal and whether the service content, charging standards, and charging items meet my expectations.”

### 3.4. Family-Level Factors

Family-level factors were reflected in five aspects: living arrangements, marital status, distance to adult children, family members’ attitudes, and informal support from family. These five aspects influenced one another and were inseparable. For example, most older people did not live alone but lived with their spouses or children, or even lived with three generations, including their spouses, children, and grandchildren. They received informal support from their spouses or children and provided some informal assistance to their children and grandchildren (e.g., taking their grandchildren to and from school). These older people generally had a lower need for CBAS integrated with medical care. Y19 was a typical example. He lived with his wife and youngest daughter. He usually took care of his grandson for his daughter while she was at work and took on the task of cooking at home. He said that he had no time and no need for CBAS. Moreover, among those who interacted with their children more frequently and received more support from their children, their children were usually opposed to their parents living in nursing homes. L01, who lived only one street away from her son, mentioned more than once that her son was very nice to her. She recalled, “At that time, I went to the nursing home and paid a little money, but when I came back home and told them (her adult children), they were opposed to me living in a nursing home.” Therefore, she needed only spiritual comfort from CBAS, and she wanted to participate in only some cultural and entertainment activities in the community.

M15, whose wife had died and whose son worked somewhere else, truly wanted to live in the community. With no one to take care of him, he had no choice but to choose a nursing home. He said, “I need (CBAS integrated with medical care). After all, my son does not live with me. Living in the community, we are familiar with each other, so I can play chess with other older people…but the community where I live does not have CBAS.” Such older people who lacked a partner or children living near them were forced to choose a nursing home. In fact, they truly needed CBAS integrated with medical care.

### 3.5. Community-Level Factors

Poor infrastructural facilities in the community made older people feel that their communities were not suitable for ageing, so they chose nursing care at home. L13 complained many times that the community where she lived did not have elevators and did not have any community staff to visit. However, poor infrastructure also contributed to increased demand among older people. L02 also complained, “Our community is not good.” She believed that the community needed to provide CBAS integrated with medical care. Some older people were not satisfied with the doctors and staff in the community. L05 suggested that the community should be equipped with more professional doctors in the future, and L04 also stressed that “ (Community staff) have to work harder and to care about what is needed for older people.”

Older people were also concerned about service fees. Since most older people had a small retirement salary and were unwilling to increase the financial burden on their children, they expressed dissatisfaction with the known excessive prices and unknown price concerns. S10 often needed rehabilitation in the community because of lumbar spine pain. She said, “The community (medical service cost) is very expensive now, (price) has increased from 200 yuan to 800 yuan.” The increasing price of services at the community hospital deterred her from purchasing medical services in the community. L02 also reflected, “The price of community services is high now we have to buy medicine every month, and it is slightly tight with our monthly retirement salary.” Moreover, they hoped that the community would increase the number of unpaid services for older people.

### 3.6. Exogenous Variables

#### 3.6.1. Gender

According to the matrix coding performed in NVivo 12.0, the 12 direct factors influencing older adults’ demand for CBAS differed by gender. Compared with females, males focused less on service fees, family members’ attitudes, informal support from families, marital status, and distance to adult children. Individuals of both sexes focused on experiences, wellness, personal opinions, community situation, and economic capability (as shown in Figure A1).

#### 3.6.2. Number of Children

The matrix coding performed in NVivo 12.0 revealed that the number of children differentially affected the factors influencing older adults’ demand for CBAS. Older people with only one adult child (one son or daughter) focused more on the distance to their adult children when considering their life after retirement. They paid more attention to service fees, individual opinion, marital status, wellness, economic capability, living conditions, community situation, and willingness to pay but cared less about informal family support than older people with more than one child. Older people with more than one child placed a greater emphasis on experiences and wellness (as shown in Figure A2).

## 4. Discussion

In this study, we identified that there is currently a relatively low demand for CBAS integrated with medical care, which is consistent with the findings of other scholars [8,25]. Although our results suggest that most older people do not need CBAS integrated with medical care, it is clear that some do not need it at present because they are not aware of it, it is not available in the community, or the cost of some rehabilitation nursing services is too high for them to afford. Currently, independent older adults care more about daily care and disease prevention, which is consistent with Deng et al.’s [29] conclusion that “people’s demands for integrated eldercare services with medical care mainly focus on daily task assistance and healthcare services”.

Moreover, based on the qualitative inductive content analysis method, all the factors influencing older adults’ demand for CBAS integrated with medical care were summarized to form a new theoretical model (shown in Figure A3): the three core factors (individual, family, and community factors) and their sub-factors, as well as two exogenous variables (the number of children and gender).

### 4.1. Individual-Level Factors

According to the analysis results, the individual category included the most reference nodes (203 tree nodes) among all categories, so formal caregivers must capture older adults’ individual information to provide comprehensive nursing support. Based on the sub-category of wellness and economic capability, older people with poor health or better economic capability needed community care services more than those with better health or poor economic capability, which is consistent with the findings of a sample of 7320 older people in Lanzhou, China [41]. Previous studies have also shown that older people with greater financial adequacy or with chronic conditions have a higher demand for CBAS [42,43]. Therefore, reducing the out-of-pocket expenses of care services would be helpful in extending the service delivery mode of CBAS integrated with medical care.

Despite the well-known benefits of CBAS, the literature continues to report that older people and their caregivers underutilize these services [44]. Underutilization of services reflects unmet care needs due to difficult access and/or low awareness of service needs by older people and medical professionals [45]. In this study, among the factor of individual opinion, fourteen older adults, more than half of the research participants, had never heard of adult services integrated with medical care, implying that the awareness of older adults regarding ageing with the support of integrated adult services and medical care in Changsha remains low. Consistent with previous empirical studies from China, the more aware older adults are of adult services integrated with medical care, the greater the demand for such services is likely to be [25,41]. Integrated care is a new concept for older Chinese people, and improving awareness of integrated care in older adults to meet their care needs is essential and urgent. Since a person’s self-awareness and experience of ageing depend on the social–spatial environment in which they live [46], we have found that some older people living in nursing homes do not truly want to live in an institution but have no one to care for them at home or in the community. However, some older adults wanted to live in a nursing home because of bad experiences in their former community. Unsurprisingly (but nevertheless disappointing), CBAS integrated with medical care in China is still not mature enough and is unsystematic, leaving older people not only unaware of such services but also not receiving the appropriate community care, which lowers the expectations of those already in need. Healthcare policy makers at the national level and communities should apply multiple approaches and organize integrated care and public health campaigns to raise older people’s awareness of the concept to help them gain knowledge of the policy, clear up misunderstandings, and remove stereotypes to better design older adult-centered service programs that reflect older people’s real needs.

### 4.2. Family-Level Factors

Previous studies have mostly highlighted family factors as an important factor influencing the integrated care of older people [10]. Interestingly, the influence of the family on older people is the result of two opposite outcomes. On the one hand, society in ancient China was based on blood ties. It is inevitable that many older people place family first, which leads to a preference for living in the community and at home. Living with adult children is still the major trend for retired older Chinese adults [47]. Older people who live with their children and receive informal support from their children tend to reinforce their children’s support through intergenerational care for their grandchildren [48]. Therefore, among the factor of living arrangements and informal support from family, some of the more able-bodied older people currently prefer to spend their time providing intergenerational support to their children or grandchildren rather than enjoying adult services in the community. However, this does not mean that such older people will not require community care in the future, and the current thinking is still based on their relatively good health. On the other hand, our research found that most older people who age alone without supportive community care service programs or family support have no choice but to age in nursing homes. Obviously, older people may be unnecessarily institutionalized in nursing homes, so older people living alone and older-only couples were more likely than co-residing households to use community-based services [49]. However, undeveloped legislation and policy systems, the gap between policy planning and implementation, the lack of service providers, and incomplete service networks are barriers to making services truly accessible [50].

At the same time, our research also found that older people’s need for CBAS with medical care services is also influenced by the attitudes of their spouse or children. If an older person still has a spouse rather than being solitary, they will usually take into account the views of their significant other when ageing in the community or a nursing home, and they will prefer to be served in the community rather than in a nursing home if their children have committed to visiting them regularly in the future or reject to them ageing in a facility. Research from Shanxi Province, China, also proves that influenced by the traditional Chinese concept of “raising children for old age” (*Yang Er Fang Lao*, a part of the Confucian cultural sphere in East Asia), children’s attitudes play a key role in the choice of older people’s retirement options, and the more supportive their children are, the more they agree with adult services with medical care, and the more willing they are to participate in them [51]. The concept of integrating CBAS with medical care should be popularized among young people with older care burdens to ensure that older people can live in a familiar community environment instead of being forced to live in a nursing home.

### 4.3. Community-Level Factors

The community situation, including infrastructural facilities in the community, service quality and other issues, is the focus of older people and a determining factor of their demand for CBAS integrated with medical care. Many older people complain that their communities do not provide appropriate and satisfactory nursing services, and they even feel that the existing community infrastructure and the level of medical and nursing staff are not sufficient for them to trust. Additionally, many older people are concerned that the cost of CBAS integrated with medical care will be excessive or unreasonable, which will be linked to and combined with individual factors such as “economic capability” and “willingness to pay” to influence the demand for services. Scholars from China have also pointed out that the development of CBAS integrated with medical care has relatively poor facilities, low quality services, and unreasonable price charges, making it difficult to meet the needs of the disabled and semi-disabled older groups [52]. The capability of a community to develop such services, the percentage of the older population in the community, and their needs should all be the focus of the community’s work. Older adults’ residential environment is associated with their health status and community-based service utilization [53]. Therefore, as the main character of service providers, the government should build relevant facilities in communities for older adults ageing in place and encourage social and non-profit organizations to be service providers in communities to create an ageing-friendly environment. For example, developing congregate meals and group activity programs to promote communication and interaction between older adults in the community will contribute to satisfying older adults’ needs and create a culture of CBAS integrated with medical care and a better community environment. Furthermore, in order to improve the medical care quality in the community, relying on contract-based family physician services, focusing on health promotion activities that prevent older adults from becoming sick, controlling and maintaining the slow development of chronic disease, and reducing unnecessary waste of medical care and eldercare resources are fundamental solutions. In addition, it is necessary to improve the pricing policy for CBAS integrated with medical care, establish a price management mechanism, reasonably assess costs and set fee standards.

### 4.4. Exogenous Variables

Our study revealed that gender and the number of children influenced older adults’ demand for eldercare services integrated with medical care to various extents. First, males focused more on personal opinions than on family members’ attitudes or relations, which is consistent with other scholars’ research that females rely more on family than males [54]. Our study also found that female older people are more concerned about the distance to their children than men. For the older people living alone in cities, the living distance of their children has a significant impact on their loneliness, and females are more sensitive than men [55]. Individuals of both sexes paid attention to experiences and wellness but to different extents. Second, the number of children influences how aware older adults are of the concept of CBAS integrated with medical care [25]. Older people with only one child showed more concern about multiple factors than older adults with more than one child. This may be because older people with fewer children have a sense of loneliness when their children are not nearby, so they focus more on retirement lifestyle and planning. Correspondingly, older adults with only one child have less informal support from family than older people with more than one child. Therefore, service providers should consider the gender of older people to provide personalized services. Additionally, individual-centered comprehensive care must be provided for older people who feel lonely or even have mental issues because their children are not around [56].

### 4.5. Comparison of Theoretical Model

Some connections and differences could be found between the proposing theoretical model of the factors influencing the demand for CBAS integrated with medical care and the Andersen’s behavioral model of health service utilization (ABMHSU) [57], which is widely used to study the influencing factors of individual medical and health care service utilization, involving older people care and medical service use, quality of life, medical expenditure, chronic disease screening and other aspects [58,59].

Both models focus on the impact of environmental and personal characteristics on the needs of people for integrated health care services, highlighting three types of factors: individual, family, and community, and suggesting different dimensions and factors along which policy makers and health service researchers can optimize health service utilization behavior and transform health service systems. Meanwhile, there are four differences between the two models. First, the theoretical model of this study takes the demand of older adults for CBAS integrated with medical care as a result, and the ABMHSU regards the demand as a core factor. Second, the theoretical model of this study has advantage in effectively identifying and controlling subtle behavior influencing factors, as it uses the number of children and gender as exogenous variables whereas the ABMHSU takes the two as sub-factors of the core factor “personal characteristics”. Third, an important innovation of this theoretical model compared to the ABMHSU is that it adds two sub-factors—“individual opinion” and “experiences” to the individual factors. Individual concepts and cognition such as “raising children for old age” (*Yang Er Fang Lao*), make the model more in line with the cultural context of China and East Asia. Fourth, ABMHSU is applicable to a wider range of scenarios and populations, but the theoretical model in this study can only be used to analyze the factors influencing the integrated care for older people in the community.

The limitation of the study might be the neglect of the views of formal caregivers (referring to community aged care services providers/staff) and informal caregivers (referring to family members). Some older people with dementia or disability often cannot express their demands independently and comprehensively, but their caregivers can reflect their needs to some extent. In addition, the cross-sectional study design is also a limitation of this study. With the gradual establishment and implementation of CBAS integrated with medical care across China, we will add the perspective of the caregivers to future studies to gain a more comprehensive understanding of the needs of older people. And the influence mechanism of the older people on the demand for the integrated care services could be further clarified through longitudinal quantitative studies in the future.

## 5. Conclusions

The adoption of a pilot qualitative study provided a novel perspective for our research, and the new topics continuously emerging from the interviews enabled us to understand older adults’ needs and feedback. Whether older people need CBAS integrated with medical care is primarily determined by personal factors, but family and community situations also play a role. As a pilot city of CBAS integrated with medical care, Changsha has made some progress in promoting the service, but it still needs to do more to meet the demands of the older adults. Since integrated care is a new thing for older Chinese people, improving the awareness of integrated care in older adults is essential and urgent. It is incumbent upon community service providers and policy makers to work together to improve the fit to increase ageing-in-place opportunities. Furthermore, this pilot study provides new knowledge of gerontological nursing by showing the characteristics of older people’s demands and the influencing factors of community-based integrated care in resource-limited settings such as Changsha, Hunan Province, China.

## Figures and Tables

**Table 1 ijerph-19-14869-t001:** Description of the CBAS integrated with medical care in China.

Dimension	Content
Service Provider	Governments, companies, healthcare institutions, NGOs (non-government organizations) or NPOs (non-profit organizations)
Modalities of Funding	Government funding, national welfare lottery, social capital or PPP (public-private partnership) mode
Service Recipients	Older people living permanently in the community or at home
Form of Service	Serve within the home- and community-based service centers, on-site service
Service Category	Daily care, primary care, rehabilitation, ancillary, psycho-spiritual support, social participation, health education, welfare and aid, etc.
Related Policies	e.g., “Notice on the Identification of the First Batch of State-level Pilot Units for ‘*Yiyang Jiehe*’ (National Health Office Family Letter [2016] No. 644)” [19], “Outline of the ‘Healthy China 2030’ Plan [2016]” [20], “Guidance on Further Promoting the Development of ‘*Yiyang Jiehe*’ (National Health Aging Issue (2022) No. 25)” [21], “Notice on Community’s ‘*Yiyang Jiehe’* Capacity Enhancement Initiative(National Health Letter on Ageing [2022] No. 53)” [22]

**Table 2 ijerph-19-14869-t002:** A summary of the sample characteristics.

ID Number	Gender	Age	Marital Status	Monthly Income	Number of Children (Son/Daughter)
L01	F	73	widowed	USD 357–571	2 (1/1)
L02	F	71	married	USD 786–841	1 (1/0)
L03	M	67	married	USD 42–143	2 (2/0)
L04	F	65	widowed	USD 357–571	1 (1/0)
L05	F	73	married	USD 357–571	3 (1/2)
Y06	M	70	married	USD 571–786	2 (0/2)
L07	M	75	married	USD 571–786	1 (1/0)
X08	F	75	married	USD 571–786	2 (0/2)
H09	F	79	widowed	USD 42–143	3 (1/2)
S10	F	65	married	USD 42–143	2 (1/1)
W11	F	69	married	USD 571–786	2 (1/1)
L12	F	73	married	USD 357–571	2 (0/2)
L13	F	65	married	USD 357–571	2 (2/0)
L14	M	88	married	USD 786–841	3 (1/2)
M15	M	72	widowed	USD 357–571	1 (1/0)
W16	F	83	widowed	USD 786–841	2 (1/1)
X17	F	89	widowed	USD 786–841	1 (0/1)
P18	M	68	married	USD 42–143	1 (1/0)
Y19	M	70	married	USD 42–143	2 (0/2)
X20	M	78	married	USD 571–786	0

**Table 3 ijerph-19-14869-t003:** Qualitative content analysis process.

Number of Coding Reference Meaning Unit	Codes	Sub-Category	Category	Theme
51	A1 Self-care ability, A2 Response actions to disease, A3 Affordability of healthcare and disease burden, A4 Chronic diseases and conditions	B1 Wellness	Individual factors	Factors influencing the demand for CBAS integrated with medical care
32	A5 Monthly income, A6 Payment for healthcare expenses	B2 Economic capability		
41	A7 Views on ageing, A8 Expectations of retirement life, A9 Opinions about providers of CBAS integrated with medical care, A10 Concern for future health, A11 Concerns about food and housing	B3 Individual opinion		
63	A12 Living experience in a nursing home, A13 Experiences with informal organizations, A14 Other people’s influence, A15 Influence of advertising campaigns, A16 Knowledge of CBAS integrated with medical care, A17 Personal career experience	B4 Experiences		
16	A18 Willing to buy eldercare services, A19 Not willing to buy eldercare services, A20 It depends	B5 Willingness to pay		
15	A21 Living with spouse, A22 Living with adult children, A23 Living with spouse and children, A24 Living in a nursing home	B6 Living arrangements	Family factors	
15	A25 Married, A26 Single	B7 Marital status		
8	A27 Lacking adult children’s support	B8 Distance to adult children		
3	A28 Spouse agrees to age in a facility, A29 Adult children refuse to allow parents to age in a facility	B9 Family Members’ Attitudes		
6	A30 Taking care of spouse, A31 Taking care of younger family members (grandchildren, children), A32 Adult children carry out filial duties	B10 Informal support from family		
31	A33 Infrastructural facilities in the community, A34 Medical quality in the community, A35 Service quality of community committee officers, A36 Performance of service providers	B11 Community situation	Community factors	
7	A37 High fees for some eldercare services, A38 Concerns regarding unknown pricing	B12 Service fees		

**Table 4 ijerph-19-14869-t004:** The factors influencing the demand for CBAS integrated with medical care.

Category	Sub-Category
Individual-level factors	Wellness
	Economic capability
	Individual opinion
	Experiences
	Willingness to pay
Family-level factors	Living arrangements
	Marital status
	Distance to adult children
	Family members’ attitude
	Informal support from family
Community-level factors	Community situation
	Service fees

## Data Availability

Not applicable.

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
