# Peer review of "Older Adults’ Demand for Community-Based Adult Services (CBAS) Integrated with Medical Care and Its Influencing Factors: A Pilot Qualitative Study in China"

_ijerph, 2022, doi:10.3390/ijerph192214869_

Round 1

Reviewer 1 Report

I reviewed the paper buy a Wang,  Wei and Liu,  and  I believe it has merit and original  content that is instructive and useful . My immediate comment is that it should be titled a pilot study given the small sample size and the lack of representation from a broader set of respondents. Having said that, there is a great need to better understand the way that community-based adult services will be deployed and accepted in China. The definition of community-based adult services (CBAS) definition that begins on line 46 could be improved if it gave a listing of exactly, according to the government,  what those CBAS include. A chart would be helpful.

 The methods seem appropriate and the qualitative descriptive design makes sense in this work. I defer to other reviewers who are currently doing qualitative research. Online from 114 it would help if there was a clear description of the two communities at different levels of economic development. I'm unable to determine what the two communities are. The inclusion criteria noted on line 123 to 126  excludes most people who are likely to need nursing homes and excludes people with dementia if I understand this.  Please provide clarification. Why exclude them? They need care the most.

A major issue which you've already noted is that there are only eight men and 12 women. That's why this a pilot study. On line 135 you note that three individuals were not covered with health insurance. Does that mean that there's no government or private insurance available to these three individuals and  why? On page 5 , table 2  gives examples from the analysis process. Can the authors be more specific about how they chose these examples? Under results, the authors refer to daily care services  but I don't know what basic medical services are. Please add a definition.

 It was helpful to have individual level factors listed followed by family level factors and community level factors. That breakout was useful. Online 306 the phrase “ choose an older people care institution” is confusing to me. Is that the same as a nursing home?

The discussion section has content on lines 359 through 367 that I believe should be moved to the literature review. Online 374 through 377, the authors stated that the “more aware older adults are of adult services integrated with medical care the greater the demand for such services will be”. That's an empirical question and can't be answered at this time based on this study.

On line 422,  the authors  state that the government should build relevant software and hardware in communities and I'm not sure what that means. Please add examples of what that would be in how it would help the older adult and the family. On line  456 the authors begins incorporating Anderson's behavioral model and I believe that does not flow. If that model was the basis of this study, it should be stated in the introductory section. If not,  further elaboration about what that model has to do with the discussion needs to be clarified.

I also believe that line 462 to 476 should be moved earlier. On line 484 the authors state that China is unsystematic, immature and detached from the real needs of older people. That would require data to state that.  Online 503 Institutional Review Board statement is said not applicable and I am confirming that that is correct. Thank you for the opportunity to review this paper.

Author Response

Dear reviewer,

Thank you for your letter and the comments concerning our manuscript. A revised version is enclosed, please see the attachment.

Best regards.

Reviewer 2 Report

The study to explore the awareness and needs  of older adults for integrated care services is worthwhile. The use of qualitative inquiry is appropriate for person-centric analysis.   The relatively minor improvements to the manuscript are as follows.

1.       The Abstract must say specifically what the awareness and need levels were. It does not tell readers anything to say the needs were “complex and multifaceted” 9as on line 22). Similarly the succeeding lines (23-24) must state the specific needs rather than simply use a blanket statement which tells readers next to nothing as to the statement justified by the findings.

2.       Introduction. Authors should end the section with a statement of their couple of research questions, rather than just implying the questions from the study aims.

3.       Punctuation and wording issues.  Authors should consider NOT to use semi-colons, where they can better just compose a new sentence (e,g, on lines 31, 33, 111, 115, 129). Moreover, authors should avoid using the same word twice in the same sentence as they do with the word integrated (e.g., lines 11- 12; 46-47).

4.       Data collection.  Under data collection, authors should state how they ensured the credibility and trustworthiness of the data.

5.       Data ANALYSIS.  Authors should edit out the reference to trustworthiness and credibility as that belongs to the data collection section. They can keep the theme dependability and confirmability in the data analysis section.

6.       Results. Authors should consider to add a Table on key themes and subthemes for a quick view of the findings.

7.       Discussion. Authors must state and discuss each of the key findings of the study. As is, they rush into commentary with no statement of the finding they are discussing. As a good example to follow is their statement and discission on gender and family composition findings as on page 11, lines 426-443. Most of the content they have under Discussion could be moved to a new section on Study Implications. That would be true of lines 359-424

Overall a great study and that would benefit from the changes indicated here.

Author Response

(The authors gave the same response as above.)

Reviewer 3 Report

This is a well written and conducted study.  I have two comments you must address:

1. Ethical issues: please do NOT identify exactly where you did the study or recruited the participants.  Instead a) give the general location in China, such as south central China; b) omit lines 116-119 which specify where you conducted this study and just keep with line 114 and 115; c), do NOT use any names to protect the identity of the participants, just use the ID number you have in Table 1.  The reason is that you have sufficient data in your table to and Results sections to identify exactly who the participant is, especially if you use the person's name or initials in ANY form.

2. Under limitations page 12 line 492, a small sample size is not a limitation for a qualitative study since your goal is to understand not generalize but explore and understand.  Now that you have draft model, you can begin to test it, develop tools to measure the concepts within it.

Author Response

(The authors gave the same response as above.)
